# RRRA: Resampling and Reranking through a Retriever Adapter

### Abstract

In dense retrieval, effective training hinges on selecting high-quality hard negatives while avoiding false negatives. Recent methods apply heuristics based on positive document scores to identify hard negatives, improving both performance and interpretability. However, these global, example-agnostic strategies often miss instance-specific false negatives. To address this, we propose a learnable adapter module that monitors Bi-Encoder representations to estimate the likelihood that a hard negative is actually a false negative. This probability is modeled dynamically and contextually, enabling fine-grained, query-specific judgments. The predicted scores are used in two downstream components: (1) resampling, where negatives are rewei-ghted during training, and (2) reranking, where top-$k$ retrieved documents are reordered at inference. Empirical results on standard benchmarks show that our adapter-enhanced framework consistently outperforms strong Bi-Encoder baselines, underscoring the benefit of explicit false negative modeling in dense retrieval.

## 1 Introduction

Dense retrieval matches queries and documents using vector similarity Karpukhin et al. (2020b), with performance hinging on informative hard negatives—non-relevant but semantically close documents Faghri et al. (2017); Robinson et al. (2020). While such negatives sharpen decision boundaries He et al. (2020); Chen et al. (2020), false negatives—relevant documents mislabeled as negatives—cause conflicting supervision and harm optimization Schroff et al. (2015); Chuang et al. (2020).

Recent self-mining methods select top-$k$ candidates by similarity Xiong et al. (2020); Zhou et al. (2022), increasing difficulty but also false negative risk under incomplete labels Qu et al. (2020). Heuristic filters such as ADORE and SimANS mitigate this with global thresholds Zhan et al. (2021a); Zhou et al. (2022), but overlook query-specific variation and may discard useful samples Ren et al. (2021a).

We propose a lightweight adapter that estimates false negative probability from Bi-Encoder representations, guiding negative reweighting during training and reranking at inference. Gradient analyses show that false negatives exhibit distinct patterns Chuang et al. (2020). Acting as a query-aware correction layer, the adapter offers cross-encoder–like signals Ren et al. (2021b) while preserving dual-encoder efficiency Luan et al. (2021).

Experiments on benchmarks such as DPR demonstrate consistent improvements over strong baselines Karpukhin et al. (2020b); Zhou et al. (2022); Ren et al. (2021a), showing that our framework provides a simple and effective solution to the false negative challenge in dense retrieval.

## 2 Related Work

### 2.1 Retrievers

Traditional methods like TF-IDF and BM25 depend on lexical overlap Yang et al. (2017); Croft et al. (2010), limiting semantic coverage. Dense retrieval instead encodes queries and documents into dense vectors using PLMs Gao & Callan (2021), achieving better semantic matching Wei et al. (2024). Models such as ColBERT reduce inference cost via late interaction Khattab & Zaharia (2020). DPR trains with in-batch negatives Karpukhin et al. (2020b), but these lack diversity Xiong

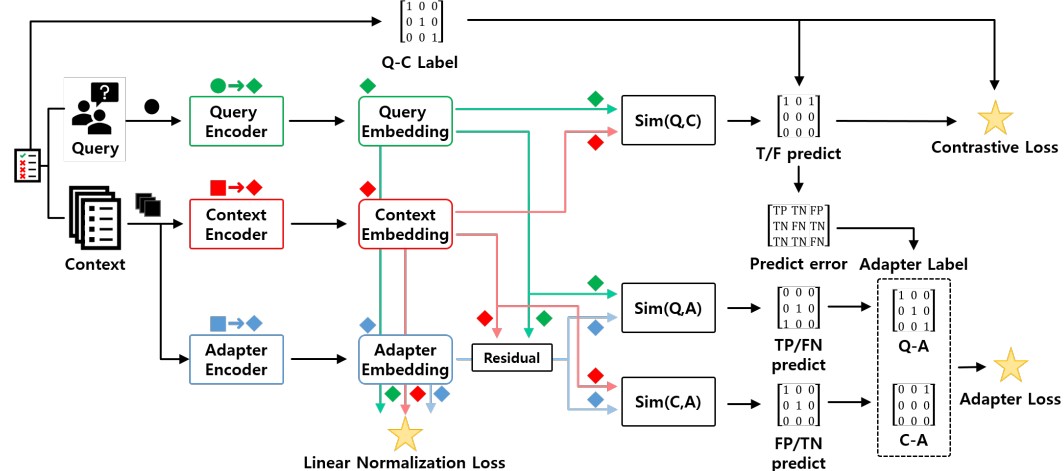

Figure 1: Architecture of the proposed RRRA framework.

et al. (2020); random negatives are often trivial Zhan et al. (2021a), and false negatives—frequent in MS MARCO—distort learning Chuang et al. (2020); Qu et al. (2020).

## 2.2 RERANKING METHODS

Cross-encoders offer strong reranking by modeling fine-grained interactions Nogueira et al. (2019); Ren et al. (2021b), but are costly. Joint training (e.g., RocketQAv2, AR2) combines retrieval and ranking losses Ren et al. (2021b;a), while SimANS and RocketQA apply soft reranking with dual encoders Zhou et al. (2022); Qu et al. (2020), though still weaker than cross-encoders.

## 2.3 NEGATIVE SAMPLING

Negative sampling is central in contrastive learning. Random or in-batch negatives Karpukhin et al. (2020b); Zhan et al. (2021a) are simple but weak; ANCE refreshes negatives during training Xiong et al. (2020), and ANCE-Tele diversifies them Sun et al. (2022), yet false negatives remain problematic Chuang et al. (2020). RocketQA mitigates this with cross-encoder filtering Qu et al. (2020), at high cost. Learnable samplers such as ADORE Zhan et al. (2021a), SimANS Zhou et al. (2022), and TriSampler Ren et al. (2021a) adapt sampling or add geometric constraints, improving informativeness. These works reflect a shift from static heuristics to adaptive, context-aware sampling, though most still lack explicit false negative handling at inference.

## 3 METHOD OVERVIEW

Figure 1 illustrates our framework, which combines: (i) Bi-Encoder contrastive learning, (ii) an Adapter for false-negative detection, (iii) integration via residual connections with optional normalization, (iv) stage-wise training (pretraining, adapter training, joint fine-tuning), and (v) resampling/re-ranking for improved negatives and inference precision.

Because multiple embedding spaces (query, document, adapter) interact, we adopt **pointwise BCE loss** for stable supervision instead of InfoNCE.

## 4 REPRESENTATION FRAMEWORK

### 4.1 CONTRASTIVE LEARNING WITH BI-ENCODER

We use a BERT-based *Bi-Encoder* retriever Karpukhin et al. (2020a), where query and document encoders share weights and produce embeddings:

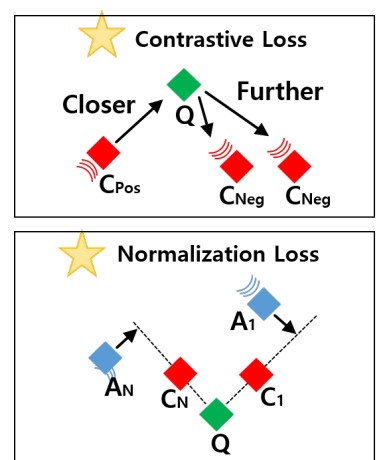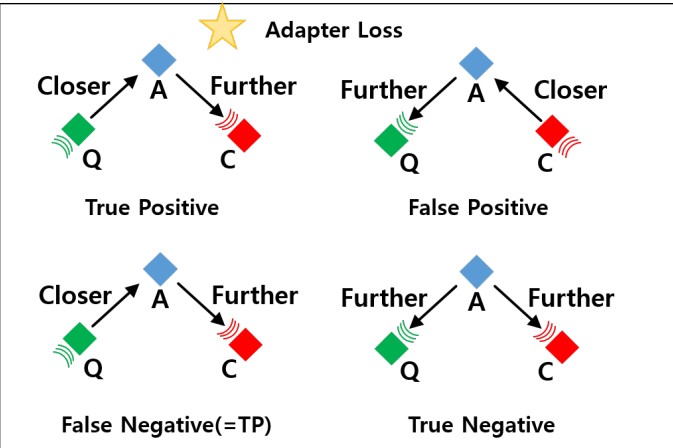

Figure 2: Label distribution predicted by the RRRA adapter.

$$\mathbf{q} = f_q(q), \quad \mathbf{d} = f_d(d).$$

Embeddings are average-pooled, and similarity is computed by cosine or dot product Ren & et al. (2021):

$$s(q, d) = \mathrm{sim}(\mathbf{q}, \mathbf{d}).$$

Training follows the *in-batch negatives* strategy Xiong & et al. (2020b), where each query's paired document is positive and all other documents in the batch are negatives. Optimization uses AdamW with linear warm-up and shuffling Gao & et al. (2021); Loshchilov & Hutter (2017).

The loss is pointwise BCE over in-batch pairs:

$$\mathcal{L}_{\mathrm{contrast}} = -\frac{1}{B} \sum_{i=1}^{B} \Big[ \log \sigma\big(s(q_i, d_i^+)\big) + \sum_{j=1, j\neq i}^{B} \log\big(1 - \sigma(s(q_i, d_j^-))\big) \Big],$$

where $B$ is batch size, $d_i^+$ the positive, and $d_j^-$ negatives. As shown in Figure 1, this provides binary supervision for each query–context pair, and Figure 2 illustrates the label distribution predicted by the RRRA Adapter under this loss.

## 4.2 ADAPTER-BASED ERROR DETECTION TASK

In Stage 2, the Bi-Encoder is frozen and only the Adapter is updated with **pointwise BCE losses** over two relations: (1) Adapter–Query and (2) Adapter–Context, using supervision derived from the Bi-Encoder's prediction outcomes.

The Adapter, initialized from the context encoder, receives context embeddings and produces a residual vector $\Delta\mathbf{d}$ to adjust the document embedding:

$$\mathbf{d}_{\mathrm{adapted}} = \mathbf{d} + \Delta\mathbf{d},$$

allowing soft correction while preserving the semantic space.

Supervision labels are assigned per outcome:

$$\mathrm{TP} : (\mathrm{A\!-\!Q} = 1,\ \mathrm{A\!-\!C} = 0), \quad \mathrm{FN} : (\mathrm{A\!-\!Q} = 1,\ \mathrm{A\!-\!C} = 0),$$
$$\mathrm{FP} : (\mathrm{A\!-\!Q} = 0,\ \mathrm{A\!-\!C} = 1), \quad \mathrm{TN} : (\mathrm{A\!-\!Q} = 0,\ \mathrm{A\!-\!C} = 0).$$

Thus, the Adapter is pulled toward the query in TP/FN, toward the context in FP, and away from both in TN.

Given similarity scores $s_{qa}$ and $s_{ca}$, the loss is

$$\mathcal{L}_{\text{adapter}} = \frac{1}{N} \sum_{i=1}^{N} \Big[ - \big( y_i^{(q)} \log \sigma(s_i^{(qa)}) + (1 - y_i^{(q)}) \log(1 - \sigma(s_i^{(qa)})) \big)$$

$$- \big( y_i^{(c)} \log \sigma(s_i^{(ca)}) + (1 - y_i^{(c)}) \log(1 - \sigma(s_i^{(ca)})) \big) \Big].$$

**Imbalance-Aware Weighting.** Since FN/FP are rarer than TP/TN, we reweight samples by imbalance ratio $r_{\text{imb}}$ with exponent $\rho$:

$$\mathcal{L}_{\text{adapter}}^{\text{weighted}} = \frac{1}{N} \sum_{i=1}^{N} \big[ \omega_i^{(q)} \cdot \text{BCE}(\sigma(s_i^{(qa)}), y_i^{(q)}) + \omega_i^{(c)} \cdot \text{BCE}(\sigma(s_i^{(ca)}), y_i^{(c)}) \big].$$

This amplifies gradients for FN/FP while keeping TP/TN stable. Importantly, $\rho$ differs from the re-sampling ratio $\gamma_{\text{RS}}$ and re-ranking ratio $\lambda_{\text{RR}}$.

Between epochs, re-sampling is applied to mined hard negatives with weights adjusted by Adapter predictions. As shown in Appendix A.3, this weighting is key to leveraging FN/FP. Additional experiments on *Label Assign* are reported in Appendix A.4.

### 4.3 Adapter–Retriever Integration

To improve detection of false negatives (FN) and false positives (FP), we integrate the Adapter with the retriever through two mechanisms: (1) relation-aware residual correction and (2) a linear normalization constraint.

#### 4.3.1 Relation-Aware Residual Correction

Query–context interaction features are constructed via difference, product, and summation:

$$\mathbf{z}_{ij} = \text{concat}\big( \mathbf{q}_i - \mathbf{c}_j, \ \mathbf{q}_i \odot \mathbf{c}_j, \ \mathbf{q}_i + \mathbf{c}_j \big),$$

which an MLP maps to a residual offset:

$$\Delta \mathbf{c}_{ij} = \text{MLP}(\mathbf{z}_{ij}), \qquad \mathbf{c}_j' = \mathbf{c}_j + \Delta \mathbf{c}_{ij}.$$

#### 4.3.2 Linear Normalization Constraint

To preserve retriever geometry, the adapted embedding is restricted to the query–context line:

$$\mathbf{a}_{ij} = \alpha \mathbf{q}_i + (1 - \alpha) \mathbf{c}_j, \quad \alpha \in [0, 1],$$

and deviations are penalized:

$$\mathcal{L}_{\text{norm}} = \frac{1}{B^2} \sum_{i=1}^{B} \sum_{j=1}^{B} \big\| (\mathbf{c}_{ij}' - \mathbf{q}_i) - \text{proj}_{(\mathbf{c}_j - \mathbf{q}_i)}(\mathbf{c}_{ij}' - \mathbf{q}_i) \big\|_2^2.$$

$$\mathcal{L}_{\text{adapter}}^{\text{weighted}} = \frac{1}{B^2} \sum_{i,j} \Big[ \omega_{ij}^{(q)} \cdot \text{BCE}(\sigma(s_{ij}^{(qa)}), y_{ij}^{(q)}) + \omega_{ij}^{(c)} \cdot \text{BCE}(\sigma(s_{ij}^{(ca)}), y_{ij}^{(c)}) \Big].$$

The final integration loss is

$$\mathcal{L}_{\text{int}} = \mathcal{L}_{\text{adapter}}^{\text{weighted}} + \lambda_{\text{norm}} \mathcal{L}_{\text{norm}},$$

where $\lambda_{\text{norm}}$ balances regularization and classification. See Section 5.1.2 and Table 5 for empirical analysis.

## 5 Training and Inference Strategy

### 5.1 Stage-wise Training Pipeline

Training proceeds in three phases: (1) Bi-Encoder pretraining, (2) Adapter training with frozen encoder, and (3) joint fine-tuning. All stages use in-batch negatives, gradient accumulation for large effective batch sizes, and imbalance-aware weighting.

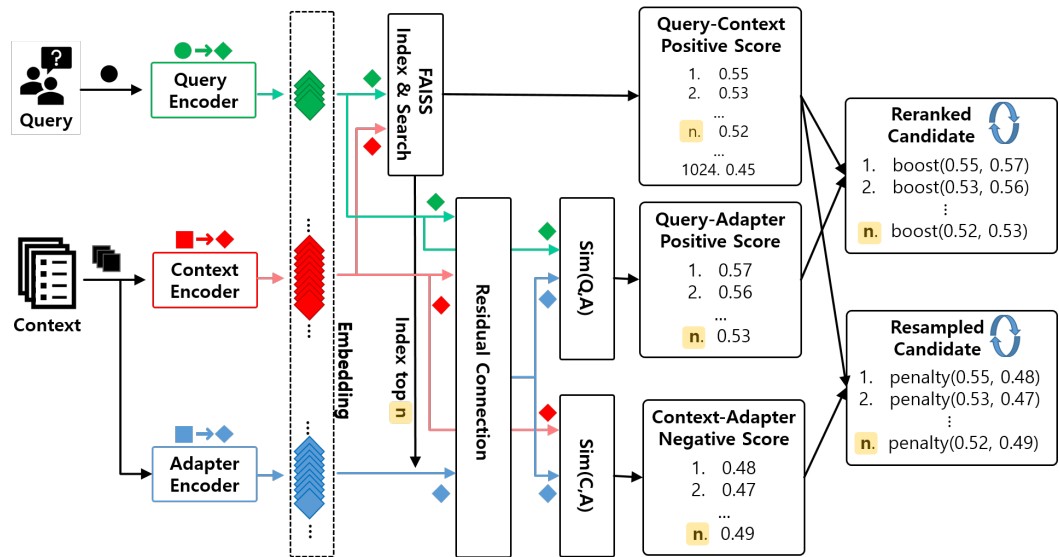

Figure 3: Effect of the ReSampling and ReRanking components.

### 5.1.1 STEP 1: BI-ENCODER PRETRAINING

A BERT-based Bi-Encoder is trained with contrastive loss and in-batch negatives to construct the base embedding space. Gradient accumulation simulates large batches, providing more diverse negatives without exceeding memory limits.

### 5.1.2 STEP 2: ADAPTER TRAINING

With the Bi-Encoder frozen, the Adapter is trained to classify query–context pairs into TP, FN, FP, or TN. Parameters are synchronized at initialization, and residual connections plus the normalization constraint stabilize learning. Imbalance-aware weighting emphasizes rare FN/FP cases, while virtual batch sizes via gradient accumulation support robust supervision. Adapter scores from this stage are later used for hard negative resampling.

### 5.1.3 STEP 3: JOINT FINE-TUNING

Both modules are fine-tuned together. Negatives include both random and mined examples, reweighted by the Adapter. Although hard negatives improve performance, they increase cost since each batch requires encoding $n + nk$ embeddings. Pointwise BCE loss provides strong pair-level supervision, and gradients are accumulated across sub-forward passes before a single update. Appendix A.7 analyzes the effect of negative counts.

**Summary.**

- **Step 1:** Bi-Encoder initialization (contrastive loss, in-batch negatives, gradient accumulation).

- **Step 2:** Adapter training (frozen encoder, residuals, normalization, imbalance-aware weighting, virtual batch expansion, error detection).

- **Step 3:** Joint training (co-optimization, hard negatives, reweighting, pointwise BCE).

Hard negatives are refreshed across epochs in Steps 2–3. Full hyperparameters are in Appendix A.1 and procedures in Appendix A.2.

| Method | NQ | | | | | TQ | | | | |
|---|---|---|---|---|---|---|---|---|---|---|
| | r@5 | r@10 | r@20 | r@50 | r@100 | r@5 | r@10 | r@20 | r@50 | r@100 |
| Bi-Encoder | 70.0 | 73.9 | 81.1 | 84.1 | 86.5 | 73.0 | 77.8 | 79.8 | 82.4 | 85.9 |
| + Random | 68.3 | 77.1 | 79.6 | 85.5 | 86.6 | 73.7 | 78.0 | 79.0 | 82.0 | 85.5 |
| + ANCE | 72.3 | 77.3 | 82.5 | 85.8 | 87.9 | 74.2 | 78.2 | 81.4 | 83.2 | 85.9 |
| + SIMANS | 75.0 | 79.5 | 84.3 | 87.0 | 89.1 | 75.2 | 81.5 | 83.9 | 85.7 | 87.1 |
| + TriSampler | 75.2 | 79.7 | 84.6 | 87.2 | 89.4 | 75.0 | 81.8 | 84.2 | 86.0 | 87.7 |
| RRRA w/o ReSampling | 74.9 | 78.3 | 82.4 | 85.8 | 88.0 | 74.9 | 80.0 | 83.3 | 84.9 | 86.6 |
| RRRA w/o ReRanking | 75.5 | 79.8 | **84.7** | 86.9 | **89.7** | 76.0 | 82.1 | **84.5** | **86.2** | **87.9** |
| RRRA (full) | **77.2** | **80.2** | **84.7** | **87.0** | 89.6 | **78.2** | **81.9** | 84.4 | 86.1 | **87.9** |

Table 1: Retrieval performance on NQ and TQ datasets.

## 5.2 SCORING FOR RE-SAMPLING AND RE-RANKING

To refine negative sampling and reduce false negatives, we use two adapter-augmented similarities: *informativeness* (QA-score, $s_{\text{HN}}$) and *false-negative likelihood* (CA-score, $s_{\text{FN}}$):

$$\text{QA-score} \equiv s_{\text{HN}}, \quad \text{CA-score} \equiv s_{\text{FN}}.$$

### 5.2.1 SCORE DEFINITIONS

For query $\mathbf{q}$, context $\mathbf{c}$, and adapter-refined embedding $\mathbf{a}$:

- **QA-score** $s_{\text{HN},i} = \text{sim}(\mathbf{q}, \mathbf{a})$: informativeness (strengthens TP/FN).
- **CA-score** $s_{\text{FN},i} = \text{sim}(\mathbf{a}, \mathbf{c})$: proxy for false-negative likelihood (suppresses FP/TN).

### 5.2.2 RE-SAMPLING (TRAINING)

From top-$k$ negatives of the frozen Bi-Encoder ($s_{\text{Base},i}$), we suppress likely FP/TN using:

$$\text{penalty}_i = |s_{\text{Base},i} - s_{\text{FN},i}|^{\gamma_{\text{RS}}}, \quad s_i^{\text{RS}} = \max\left(\frac{s_{\text{Base},i} + s_{\text{FN},i}}{2} \cdot (1 - \text{penalty}_i), 0\right),$$

where $\gamma_{\text{RS}}$ controls suppression strength (Appendix A.6).

### 5.2.3 RE-RANKING (INFERENCE)

At inference, top-100 results are refined by boosting TP/FN:

$$\text{boost}_i = \left(1 - |s_{\text{Base},i} - s_{\text{HN},i}|\right)^{\lambda_{\text{RR}}}, \quad s_i^{\text{RR}} = \max\left(\frac{s_{\text{Base},i} + s_{\text{HN},i}}{2} \cdot (1 + \text{boost}_i), 0\right),$$

where $\lambda_{\text{RR}}$ adjusts the Adapter's impact.

Since $\mathbf{c}$ and $\mathbf{a}$ can be precomputed and indexed (e.g., FAISS), QA/CA scores are obtained efficiently. The same procedure is used for both training and inference—precompute embeddings, retrieve with $\mathbf{q}$, align indices, then compute QA/CA scores for reweighting—ensuring consistent robustness and accuracy (Fig. 3). Further analyses on the choice of $\gamma_{\text{RS}}$, $\lambda_{\text{RR}}$, and the number of hard negatives are provided in Appendices A.5–A.7.

## 6 EXPERIMENTS

### 6.1 EXPERIMENTAL SETUP

**Datasets.** We evaluate on four standard benchmarks: Natural Questions (NQ) Kwiatkowski et al. (2019), TriviaQA (TQA) Joshi et al. (2017), MS MARCO Passage (MSPas), and MS MARCO Document (MSDoc) Nguyen et al. (2016). Following prior work Zhou et al. (2022); Ren et al. (2021a), we sample negatives using a 1:15 positive-to-negative ratio. For each query, we retrieve the top-1024 passages with Faiss Johnson et al. (2019) as negative candidates.

| Method | MS-Pas | | | | | MS-Doc | | | | |
|---|---|---|---|---|---|---|---|---|---|---|
| | r@10 | r@20 | r@50 | r@100 | MRR@10 | r@10 | r@20 | r@50 | r@100 | MRR@10 |
| Bi-Encoder | 55.4 | 67.6 | 80.2 | 87.9 | 34.3 | 58.3 | 69.6 | 81.4 | 86.9 | 35.4 |
| + Random | 55.3 | 67.7 | 80.9 | 89.2 | 34.1 | 60.0 | 71.5 | 82.0 | 87.7 | 34.3 |
| + ANCE | 54.9 | 67.5 | 81.3 | 89.5 | 34.2 | 61.9 | 74.7 | 85.0 | 90.0 | 35.4 |
| + SIMANS | 58.0 | 69.2 | 81.9 | 89.9 | 37.0 | 63.8 | 75.3 | 86.4 | 90.7 | 36.8 |
| + TriSampler | 58.2 | 69.7 | 82.3 | 90.9 | 37.2 | 64.0 | 75.7 | 87.0 | 91.2 | 35.6 |
| RRRA w/o ReSampling | 58.3 | 68.3 | 82.1 | 87.9 | 36.3 | 62.5 | 72.6 | 82.4 | 87.4 | 38.3 |
| RRRA w/o ReRanking | 58.8 | **69.9** | **83.8** | **91.1** | 36.9 | 64.1 | **76.6** | **87.5** | **91.7** | 36.4 |
| RRRA (full) | **59.4** | **69.9** | 83.5 | 90.4 | **38.1** | **65.7** | 76.1 | 86.9 | **91.7** | **40.8** |

Table 2: Retrieval performance on MS-Pas and MS-Doc datasets.

| Dataset | Train | Dev | Test | # Documents |
|---|---|---|---|---|
| NQ | 58,880 | 8,757 | 3,610 | 21,015,324 |
| TQA | 60,413 | 8,837 | 11,313 | 21,015,324 |
| MSPas | 502,939 | 6,980 | - | 8,841,823 |
| MSDoc | 367,013 | 5,193 | - | 3,213,835 |

Table 3: Statistics of datasets used for training and evaluation.

**Evaluation Metrics.** We report Recall@$k$ ($k \in \{5, 10, 20, 50, 100\}$) and Mean Reciprocal Rank at 10 (MRR@10), which together measure retrieval depth and top-ranked precision.

**Baselines.** We compare against random sampling, ANCE Xiong et al. (2020), SimANS Zhou et al. (2022), TriSampler Ren et al. (2021a), and external retrievers such as BM25 Yang et al. (2017), DPR Karpukhin et al. (2020b), RocketQA Qu et al. (2020), and ADORE Zhan et al. (2021a). Our method (RRRA) is evaluated in both ReSampling (training) and ReRanking (inference) modes, along with stage-wise results.

**RRRA Details.** All models use a BERT-base dual encoder backbone. Stage 2 employs *ContextE Init* for adapter initialization, and Stage 3 incorporates mined hard negatives with gradient accumulation. Key hyperparameters include imbalance weight $\gamma_{\text{imb}} = 0.3$, re-sampling ratio $\gamma_{\text{RS}}$, and re-ranking ratio $\lambda_{\text{RR}}$, tuned on the dev set.

## 6.2 RESULTS ON NQ AND TQ

Table 1 shows that on both datasets, reranking alone (*RRRA w/o ReSampling*) improves top ranks (R@5/10), resampling alone (*RRRA w/o ReRanking*) strengthens deeper recall (R@50/100), and the full RRRA balances both, delivering consistent or Pareto-competitive gains. Without requiring cross-encoders, the lightweight Adapter applies residual corrections while preserving Bi-Encoder geometry, enabling precise reranking, robust negative modeling, and overall performance that surpasses heuristic sampling and rivals or exceeds strong baselines such as ANCE, SimANS, and TriSampler.

## 6.3 RESULTS ON MS MARCO

Table 2 shows that RRRA outperforms bi-encoder–based samplers (Random, ANCE, SimANS, TriSampler) under the same backbone: on MS-Pas it improves early ranks (MRR@10, R@10/20) through query–adapter corrections, on both MS-Pas and MS-Doc it boosts tail recall (R@50/100) by suppressing misleading negatives, and on MS-Doc it remains robust thanks to the normalization constraint. Ablations further confirm that reranking mainly improves top ranks, resampling benefits deeper ones, and the full model balances both, consistently outperforming heuristic sampling.

| Method | NQ | | TQ | | MS-Pas | | MS-Doc | |
|---|---|---|---|---|---|---|---|---|
| | R@5 | R@100 | R@5 | R@100 | MRR@10 | R@50 | MRR@10 | R@100 |
| BM25 Robertson (2009) | – | 73.7 | – | 76.7 | 18.7 | 59.2 | 27.9 | 80.7 |
| DPR Karpukhin et al. (2020c) | – | 85.4 | – | 84.9 | – | – | 32.0 | 86.4 |
| ANCE Xiong & et al. (2020a) | 71.8 | 87.5 | 72.4 | 85.3 | – | 81.1 | 37.7 | 89.4 |
| RocketQA Qu & et al. (2021) | 74.0 | 88.5 | 76.1 | – | – | 85.5 | – | – |
| ADORE Zhan et al. (2021b) | – | – | – | – | – | – | 40.5 | 91.9 |
| SimANS Zhan & et al. (2022) | 78.6 | 90.3 | 78.6 | 88.1 | 40.9 | 88.7 | 43.1 | 92.3 |
| TriSampler Liu & et al. (2024) | – | 90.7 | – | 88.5 | 41.4 | 89.1 | – | 93.1 |
| RRRA w/o ReRanking | 74.9 | 88.0 | 74.9 | 86.6 | 36.3 | 82.1 | 38.3 | 87.4 |
| RRRA w/o ReSampling | 75.5 | 89.7 | 76.0 | 87.9 | 36.9 | 83.8 | 36.4 | 91.7 |
| RRRA (full) | 77.2 | 89.6 | 78.2 | 87.9 | 38.1 | 83.5 | 40.8 | 91.7 |

Table 4: Retrieval performance comparison across NQ, TQ, MS-Pas, and MS-Doc. Reported metrics follow standard cutoffs per dataset. All scores are scaled by 100 except MRR values.

## 6.4 COMPARISON WITH STRONG BASELINES

Table 4 shows that, despite using only a BERT-base bi-encoder with a lightweight Adapter, RRRA is competitive with stronger baselines such as SimANS, TriSampler, and RocketQA. Its advantage comes from explicit false-negative modeling: the QA path sharpens early ranks, the CA path improves tail recall, and ablations confirm both are necessary. Overall, RRRA achieves robust performance without cross-encoders or heavy supervision.

| Method | Macro-F1 (%) | Acc (%) | Prec. (%) | Rec. (%) | Bal. Acc (%) | Pred AUC (%) |
|---|---|---|---|---|---|---|
| Adapter w/o Residual | 63.9 | 69.0 | 65.4 | 68.2 | 69.8 | 74.1 |
| Adapter w/o Linear Norm | 85.2 | 83.6 | 80.3 | 83.2 | 84.0 | 88.2 |
| Adapter w/o FT-FN Ratio | 90.9 | 84.6 | 82.1 | 85.0 | 85.4 | 88.9 |
| Adapter w/o ContextE Init | 92.2 | 89.1 | 86.2 | 88.4 | 89.3 | 93.5 |
| **Full Adapter** | **93.3** | **89.4** | **87.1** | **89.0** | **89.3** | **94.0** |

Table 5: Comparison of adapter variants on classification performance (all metrics reported in %).

## 6.5 COMPONENT ABLATION STUDY

To assess each component's contribution, we conduct an ablation study by selectively disabling individual adapter modules and evaluating F1 score, which measures how accurately the adapter identifies false positives and false negatives during Stage 2. Table 5 shows that removing any component leads to performance degradation, confirming the importance of residual connections, normalization, ratio modeling, and initialization (ContextE Init: initializing the adapter from the context encoder before training). Experiments are conducted on the MS MARCO Document dataset with the full training set, using 5 training epochs and imbalance weighting parameter $\gamma = 0.3$. All components contribute meaningfully to the adapter's effectiveness.

## 7 GRADIENT ANALYSIS OF REWEIGHTED NEGATIVES

On MS MARCO-Doc, we inspect *normalized* document-encoder gradients during Stage 3 (parameters frozen, forward-only) by comparing *top-$k$ mining* with *RRRA resampling* over 100 queries and 1,000 candidates (Figure 4).[1] RRRA redistributes gradients more evenly: it dampens noisy spikes at the top ranks, maintains stronger signals at mid-depths, and keeps the tail stable. This acts as a curriculum-like effect, downweighting likely false negatives while emphasizing informative items, resulting in **well-regulated gradients** that improve stability and retrieval quality.

---

[1]Random, SimANS, and TriSampler show intermediate patterns; omitted for clarity.

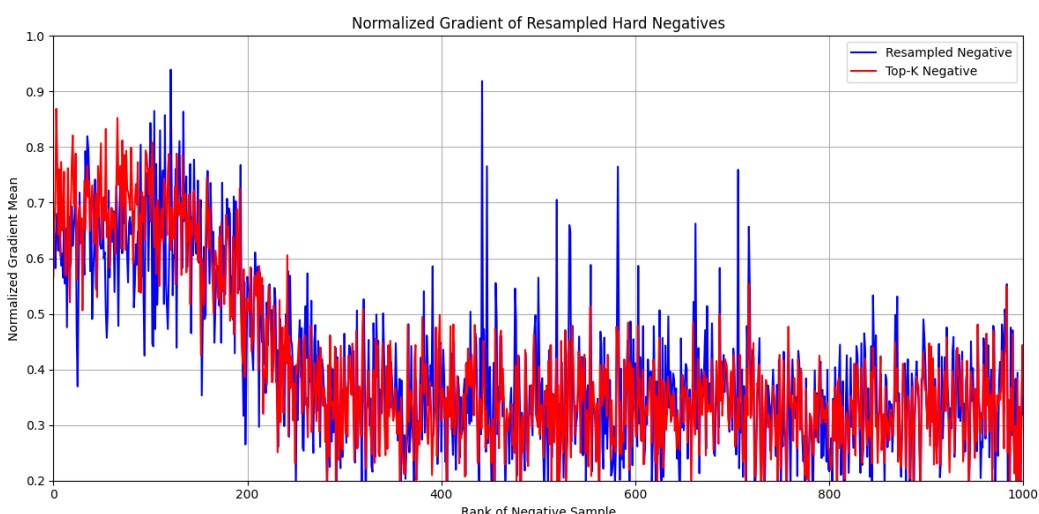

Figure 4: Normalized Gradient of Resampled Hard Negatives

# 8 CONCLUSION AND DISCUSSION

We presented **RRRA**, a dense retrieval framework that mitigates false negatives through a learnable adapter applied to both training-time resampling and inference-time reranking. Unlike heuristic, global-threshold methods, RRRA models instance-level false-negative likelihood from encoder states, enabling targeted filtering, reweighting, and reranking of top-$k$ hard negatives. By leveraging query–context interactions and residual corrections, RRRA adaptively reshapes supervision, yielding stable gradients and improving retrieval precision with minimal overhead.

Experiments on NQ, TQ, MS-Pas, and MS-Doc show consistent gains: reranking sharpens top ranks, resampling improves deeper recall, and the combined model outperforms or rivals strong baselines such as SimANS, TriSampler, and ADORE, despite using only a bi-encoder backbone. Since RRRA reranks within the bi-encoder space over a shortlist of 200 candidates, it is naturally faster than cross-encoder reranking; nonetheless, a direct speed comparison with cross-encoders is an important next step. Current limitations include reliance on the base encoder, the need for further optimization of reranking, and more principled tuning of hyperparameters such as the imbalance exponent, re-sampling ratio, and re-ranking ratio. Promising extensions include combining RRRA with cross-encoder distillation, alternative backbones, or advanced negative mining strategies, as well as out-of-domain evaluation (e.g., BEIR) and qualitative error analysis.

Overall, RRRA offers a simple, scalable, and modular approach to learning-based negative sampling, providing a foundation for subsequent work. While not exhaustive in scope, this study introduces the core concept and demonstrates its potential, leaving broader integration and optimization to future research.

### AUTHOR CONTRIBUTIONS

If you'd like to, you may include a section for author contributions as is done in many journals. This is optional and at the discretion of the authors.

### ACKNOWLEDGMENTS

Use unnumbered third level headings for the acknowledgments. All acknowledgments, including those to funding agencies, go at the end of the paper.

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
