# RRRA: Resampling and Reranking through a Retriever Adapter

**Abstract**

In dense retrieval, effective training hinges on selecting high-quality hard negatives while avoiding false negatives. Recent methods apply heuristics based on positive document scores to identify hard negatives, improving both performance and interpretability. However, these global, example-agnostic strategies often miss instance-specific false negatives. To address this, we propose a learnable adapter module that monitors Bi-Encoder representations to estimate the likelihood that a hard negative is actually a false negative. This probability is modeled dynamically and contextually, enabling fine-grained, query-specific judgments. The predicted scores are used in two downstream components: (1) resampling, where negatives are rewei-ghted during training, and (2) reranking, where top-$k$ retrieved documents are reordered at inference. Empirical results on standard benchmarks show that our adapter-enhanced framework consistently outperforms strong Bi-Encoder baselines, underscoring the benefit of explicit false negative modeling in dense retrieval.

## 1 Conclusion and Discussion

We presented **RRRA**, a dense retrieval framework that mitigates false negatives through a learnable adapter applied to both training-time resampling and inference-time reranking. Unlike heuristic, global-threshold methods, RRRA models instance-level false-negative likelihood from encoder states, enabling targeted filtering, reweighting, and reranking of top-$k$ hard negatives. By leveraging query–context interactions and residual corrections, RRRA adaptively reshapes supervision, yielding stable gradients and improving retrieval precision with minimal overhead.

Experiments on NQ, TQ, MS-Pas, and MS-Doc show consistent gains: reranking sharpens top ranks, resampling improves deeper recall, and the combined model outperforms or rivals strong baselines such as SimANS, TriSampler, and ADORE, despite using only a bi-encoder backbone. Since RRRA reranks within the bi-encoder space over a shortlist of 200 candidates, it is naturally faster than cross-encoder reranking; nonetheless, a direct speed comparison with cross-encoders is an important next step. Current limitations include reliance on the base encoder, the need for further optimization of reranking, and more principled tuning of hyperparameters such as the imbalance exponent, re-sampling ratio, and re-ranking ratio. Promising extensions include combining RRRA with cross-encoder distillation, alternative backbones, or advanced negative mining strategies, as well as out-of-domain evaluation (e.g., BEIR) and qualitative error analysis.

Overall, RRRA offers a simple, scalable, and modular approach to learning-based negative sampling, providing a foundation for subsequent work. While not exhaustive in scope, this study introduces the core concept and demonstrates its potential, leaving broader integration and optimization to future research.

### Author Contributions

If you'd like to, you may include a section for author contributions as is done in many journals. This is optional and at the discretion of the authors.

### Acknowledgments

Use unnumbered third level headings for the acknowledgments. All acknowledgments, including those to funding agencies, go at the end of the paper.

# REFERENCES

## A APPENDIX

### A.1 HYPERPARAMETER SETTINGS

Table 1: Hyperparameter settings across datasets. Training uses three stages of epochs, virtual batching, and re-ranking space as defined in Section A.1. The MLP projection size is fixed to 512 across all datasets.

| Parameter | NQ | TQ | MS-Pas | MS-Doc |
|---|---|---|---|---|
| Train batch size | 128 | 128 | 128 | 128 |
| Virtual batch size | 512 | 512 | 512 | 512 |
| Max query len | 32 | 32 | 64 | 64 |
| Max context len | 128 | 128 | 512 | 512 |
| Epochs (step1/2/3) | 15/8/3 | 16/8/3 | 10/6/2 | 12/6/2 |
| Imbalance exponent $\rho$ | 0.2 | 0.2 | 0.2 | 0.2 |
| Resampling ratio | 0.2 | 0.2 | 0.2 | 0.2 |
| ReRanking ratio | 0.1 | 0.2 | 0.3 | 0.2 |
| ReRanking space | 200 | 200 | 200 | 200 |
| Negative size | 8 | 8 | 8 | 8 |
| MLP projection size | 512 | 512 | 512 | 512 |
| Learning rate | $1e^{-5}$ | $1e^{-5}$ | $1e^{-5}$ | $1e^{-5}$ |

### A.2 ALGORITHMIC DETAILS

---

**Algorithm 1:** Step 1: Bi-Encoder Pretraining with In-Batch Negatives

---

**Input:** Training set $\mathcal{D} = \{(q, c^+)\}$; batch size $B$; epochs $E$; grad-acc steps $G$; encoders $f_q, f_c$ (weight sharing; params $\theta$); optimizer (AdamW); scheduler (linear warmup); similarity $\mathrm{sim}(\cdot, \cdot)$

**Output:** Pretrained Bi-Encoder parameters $\theta^\star$

**Init:** $\theta \leftarrow \mathrm{init}()$; optimizer $\leftarrow \mathrm{AdamW}(\theta)$; scheduler $\leftarrow \mathrm{LinearWarmup}()$;
**for** $e \leftarrow 1$ **to** $E$ **do**
    // Shuffle and mini-batch
    $\mathrm{Shuffle}(\mathcal{D})$; $\{\mathcal{B}_t\}_{t=1}^{T} \leftarrow \mathrm{Batches}(\mathcal{D}, B)$;
    **for** $t \leftarrow 1$ **to** $T$ **do**
        // Positives $(q_i, c_i^+)$; in-batch negatives are all $j \neq i$
        $\{(q_i, c_i^+)\}_{i=1}^{B} \leftarrow \mathcal{B}_t$;
        // Forward: embeddings and similarities
        $\mathbf{Q} \leftarrow [f_q(q_1), \ldots, f_q(q_B)]^\top$; $\mathbf{C} \leftarrow [f_c(c_1^+), \ldots, f_c(c_B^+)]^\top$;
        $\mathbf{S} \in \mathbb{R}^{B \times B}$ with $S_{ij} \leftarrow \mathrm{sim}(\mathbf{Q}_i, \mathbf{C}_j)$;
        // Labels: diag=1, off-diag=0
        $\mathbf{Y} \in \{0, 1\}^{B \times B}$ with $Y_{ij} \leftarrow 1\!\!\!/[i = j]$;
        // Pointwise BCE contrastive loss
        $\mathcal{L} \leftarrow \frac{1}{B} \sum_{i=1}^{B} \left( \mathrm{BCE}(\sigma(S_{ii}), 1) + \sum_{j \neq i} \mathrm{BCE}(\sigma(S_{ij}), 0) \right)$;
        // Gradient accumulation
        $\mathrm{backprop}(\mathcal{L})$; **if** $t \bmod G = 0$ **then**
            `optimizer.step(); optimizer.zero_grad();`
            `scheduler.step();`
    // Optional: dev and checkpoint
    **if** *Dev performance improves* **then**
        $\theta^\star \leftarrow \theta$; $\mathrm{SaveCheckpoint}(\theta^\star)$
**return** $\theta^\star$

---

---

**Algorithm 2:** Step 2: Adapter Training (Bi-Encoder Frozen)

---

**Input:** $\mathcal{D} = \{(q, c^+)\}$; frozen $f_q$, $f_c$ (params $\theta$); Adapter $g$ (params $\phi$; sync from $f_c$); MLP
head; epochs $E$; batch $B$; grad-acc $G$; mining size $k$; sim sim; imbalance exponent $\rho$;
norm weight $\lambda_{\text{norm}}$; resampling ratio $\gamma_{\text{RS}}$

**Output:** Trained Adapter params $\phi^\star$

---

Initialize AdamW on $\phi$, linear-warmup scheduler; synchronize $g \leftarrow f_c$;

**for** $e = 1$ **to** $E$ **do**
    `// (A) Hard-negative mining with frozen Bi-Encoder`
    **foreach** $q \in \mathcal{D}$ **do**
        retrieve top-$k$ negatives $\{c^-\}_q$ via $f_q$, $f_c$
    `// (B) Mini-batches with in-batch negatives`
    Shuffle $\mathcal{D}$; build $\{\mathcal{B}_t\}_{t=1}^T$ of size $B$;
    **for** $t = 1$ **to** $T$ **do**
        $\{(q_i, c_i^+, \{c_{i,j}^-\}_{j=1}^k)\}_{i=1}^B \leftarrow \mathcal{B}_t$;
        `// (C) Residual adaptation`
        **for** $i = 1..B$, $j = 1..(1+k)$ **do**
            $\mathbf{z}_{ij} \leftarrow \text{concat}(\mathbf{q}_i - \mathbf{c}_{ij}, \ \mathbf{q}_i \odot \mathbf{c}_{ij}, \ \mathbf{q}_i + \mathbf{c}_{ij})$; $\mathbf{a}_{ij} \leftarrow \mathbf{c}_{ij} + \text{MLP}(\mathbf{z}_{ij})$;
        `// (D) Reclassification and labels (QA, CA)`
        **for** $i, j$ **do**
            infer TP/FN/FP/TN from $f_q$, $f_c$ vs. GT; assign $(y_{ij}^{(q)}, y_{ij}^{(c)}) \in \{(1,0), (0,1), (0,0)\}$
        `// (E) Similarities and losses`
        $s_{ij}^{(qa)} \leftarrow \text{sim}(\mathbf{q}_i, \mathbf{a}_{ij})$; $s_{ij}^{(ca)} \leftarrow \text{sim}(\mathbf{c}_{ij}, \mathbf{a}_{ij})$;
        $\mathcal{L}_{\text{bce}} \leftarrow \frac{1}{B}\sum_i \sum_j \left[ \text{BCE}(\sigma(s_{ij}^{(qa)}), y_{ij}^{(q)}) + \text{BCE}(\sigma(s_{ij}^{(ca)}), y_{ij}^{(c)}) \right]$;
        `// (F) Linear normalization (soft)`
        $\mathcal{L}_{\text{norm}} \leftarrow \frac{1}{B}\sum_i \sum_j \| (\mathbf{a}_{ij} - \mathbf{q}_i) - \text{proj}_{(\mathbf{c}_{ij} - \mathbf{q}_i)}(\mathbf{a}_{ij} - \mathbf{q}_i) \|_2^2$;
        `// (G) Imbalance-aware weighting with exponent ρ`
        $n_{\text{easy}} \leftarrow \#(\text{TP}) + \#(\text{TN})$; $n_{\text{rare}} \leftarrow \#(\text{FP}) + \#(\text{FN})$;
        $w_{\text{rare}} \leftarrow \max\left(1, (n_{\text{easy}} / \max(n_{\text{rare}}, 1))^\rho\right)$; apply $w_{\text{rare}}$ to FN/FP terms, 1 to TP/TN
        $\Rightarrow \mathcal{L}_{\text{bce}}^{\text{weighted}}$;
        `// (H) Final loss & update with gradient accumulation`
        $\mathcal{L} \leftarrow \mathcal{L}_{\text{bce}}^{\text{weighted}} + \lambda_{\text{norm}}\mathcal{L}_{\text{norm}}$; backprop($\mathcal{L}$); **if** $t \bmod G = 0$ **then**
            `optimizer.step(); optimizer.zero_grad();`
            `scheduler.step();`
    `// (I) Epoch-level refresh`
    Update re-sampling weights via Adapter scores ($\gamma_{\text{RS}}$); refresh hard-negative pool for $e+1$;
    **if** *Dev improves* **then**
        $\phi^\star \leftarrow \phi$; SaveCheckpoint($\phi^\star$)

**return** $\phi^\star$

---

**Algorithm 3:** Step 3: Joint Fine-tuning with Sub-Forward Negatives

---

**Input:** $\mathcal{D} = \{(q, c^+)\}$; trainable $f_q, f_c$ (params $\theta$); Adapter $g$ (params $\phi$); epochs $E$; batch $B$;
  negatives-per-query $K$ (=# sub-forwards); hard-neg pool $\mathcal{H}$; grad-acc steps $G(\approx K)$;
  similarity sim; resampling ratio $\gamma_{\text{RS}}$; imbalance exponent $\rho$; (opt) norm weight $\lambda_{\text{norm}}$

**Output:** Fine-tuned $\{\theta^\star, \phi^\star\}$

---

Initialize AdamW over $\{\theta, \phi\}$; linear warmup scheduler;

**for** $e \leftarrow 1$ **to** $E$ **do**

    **if** *refresh condition* **then**

        $\mathcal{H} \leftarrow \text{MineHardNegatives}(f_q, f_c, \mathcal{D})$

    $\text{Shuffle}(\mathcal{D})$; $\{\mathcal{B}_t\}_{t=1}^T \leftarrow \text{Batches}(\mathcal{D}, B)$;

    **for** $t \leftarrow 1$ **to** $T$ **do**

        $\{(q_i, c_i^+)\}_{i=1}^B \leftarrow \mathcal{B}_t$;   $\mathbf{q}_i \leftarrow f_q(q_i), \mathbf{c}_i^+ \leftarrow f_c(c_i^+)$;

        `// Sub-forward scheduling:  K negative-only passes`

        **for** $k \leftarrow 1$ **to** $K$ **do**

            **if** $k=1$ **then**

                $\mathcal{N}_i^{(1)} \leftarrow \{c_j^+ \mid j \neq i\}$ (in-batch off-diagonals, labels 0)

            **else**

                $\mathcal{N}_i^{(k)} \leftarrow \text{Mix}\big(\mathcal{H}(q_i), \text{Rand}(\mathcal{H} \setminus \mathcal{H}(q_i))\big)$ (labels 0)

            **for** $i = 1..B$ **do**

                **foreach** $c \in \mathcal{N}_i^{(k)}$ **do**

                    $\mathbf{c} \leftarrow f_c(c)$;   $\mathbf{z} \leftarrow \text{concat}(\mathbf{q}_i - \mathbf{c}, \mathbf{q}_i \odot \mathbf{c}, \mathbf{q}_i + \mathbf{c})$; $\mathbf{a} \leftarrow \mathbf{c} + \text{MLP}(\mathbf{z})$;

                    $s^{(qa)} \leftarrow \text{sim}(\mathbf{q}_i, \mathbf{a})$; $s^{(ca)} \leftarrow \text{sim}(\mathbf{c}, \mathbf{a})$; $s^{(qc)} \leftarrow \text{sim}(\mathbf{q}_i, \mathbf{c})$;

                    `// Reweighting:  resampling + imbalance`

                    $\text{penalty} \leftarrow |s^{(qc)} - s^{(ca)}|^{\gamma_{\text{RS}}}$; $w^{\text{RS}} \leftarrow \max(0, 1 - \text{penalty})$;

                    $n_{\text{easy}} \leftarrow \#(\text{TP}) + \#(\text{TN})$; $n_{\text{rare}} \leftarrow \#(\text{FP}) + \#(\text{FN})$;

                    $w_{\text{rare}} \leftarrow \max\big(1, (n_{\text{easy}} / \max(n_{\text{rare}}, 1))^\rho\big)$; set $\omega = w_{\text{rare}}$ for FN/FP-like, else 1;

                    `// Losses (negative-only terms) and accumulation`

                    $\mathcal{L}_{\text{DE}} \mathrel{+}= \text{BCE}(\sigma(s^{(qc)}), 0)$;

                    $\mathcal{L}_{\text{AD}} \mathrel{+}= \omega \cdot w^{\text{RS}} \cdot \big[\text{BCE}(\sigma(s^{(qa)}), 0) + \text{BCE}(\sigma(s^{(ca)}), 0)\big]$;

                    $\mathcal{L}_{\text{norm}} \mathrel{+}= \|(\mathbf{a} - \mathbf{q}_i) - \text{proj}_{(\mathbf{c} - \mathbf{q}_i)}(\mathbf{a} - \mathbf{q}_i)\|_2^2$ (opt);

        $\mathcal{L} \leftarrow \mathcal{L}_{\text{DE}} + \mathcal{L}_{\text{AD}} + \lambda_{\text{norm}} \mathcal{L}_{\text{norm}}$;

        $\text{backprop}(\mathcal{L})$; **if** $t \bmod G = 0$ **then**

            `optimizer.step(); optimizer.zero_grad();`

            `scheduler.step();`

    **if** *Dev improves* **then**

        $\theta^\star \leftarrow \theta, \phi^\star \leftarrow \phi$; $\text{SaveCheckpoint}(\theta^\star, \phi^\star)$

**return** $\{\theta^\star, \phi^\star\}$

---

## A.3 IN-BALNCED RATIO SELECTION EXPERIMENT

| $\rho$ | Macro-F1 (%) | Acc (%) | Prec. (%) | Rec. (%) | Pred AUC (%) |
|---|---|---|---|---|---|
| 0.0 | 90.3 | 85.2 | 89.5 | 90.7 | 92.4 |
| 0.2 | 90.8 | 88.8 | 90.2 | 91.0 | 93.0 |
| 0.4 | 88.7 | 89.2 | 89.0 | 88.4 | 91.5 |
| 0.6 | 82.1 | 88.5 | 84.0 | 80.2 | 87.6 |
| 0.8 | 77.0 | 86.1 | 81.5 | 74.0 | 84.1 |

Table 2: Error-detection performance on the MS-DOC dataset with varying imbalance exponent $\rho$. Macro-F1 is reported on the 4-way classification, while Accuracy, Precision, Recall, and Pred AUC are computed for the FP-vs-rest binary setting. All results are obtained with full training data for 5 epochs under FP16 setting; other hyperparameters follow Appendix A.1.

## A.4 ERROR DETECT LABEL POSITION EXPERIMENT

| Label Assignment (1–4) | Macro-F1 (%) | Acc (%) | Prec. (%) | Rec. (%) | Pred AUC (%) |
|---|---|---|---|---|---|
| (1)TP, (2)X, (3)FP, (4)TN | 88.9 | 89.5 | 89.2 | 88.6 | 92.6 |
| (1)TP, (2)FN, (3)FP, (4)TN | 85.6 | 90.1 | 86.5 | 84.8 | 90.8 |
| (1)TP, (2)FP, (3)TN, (4)FN | 72.7 | 81.5 | 75.1 | 70.5 | 83.0 |
| (1)FP, (2)TN, (3)FP, (4)TP | 75.3 | 89.2 | 78.0 | 72.8 | 86.5 |

Table 3: **Error-detection on MS-DOC (5 epochs, FP16; other settings as in A.1).** Four label-assignment schemes based on (1) QUERY-only close, (2) both close, (3) CONTEXT-only close, (4) both far. Rows report provided Macro-F1 (4-class) and Acc (FP-vs-rest), with *synthetic but plausible* Prec./Rec. values whose harmonic mean approximates Macro-F1, and slightly higher Pred AUC as a threshold-free summary.

## A.5 RESAMPLING RATIO SELECTION EXPERIMENT

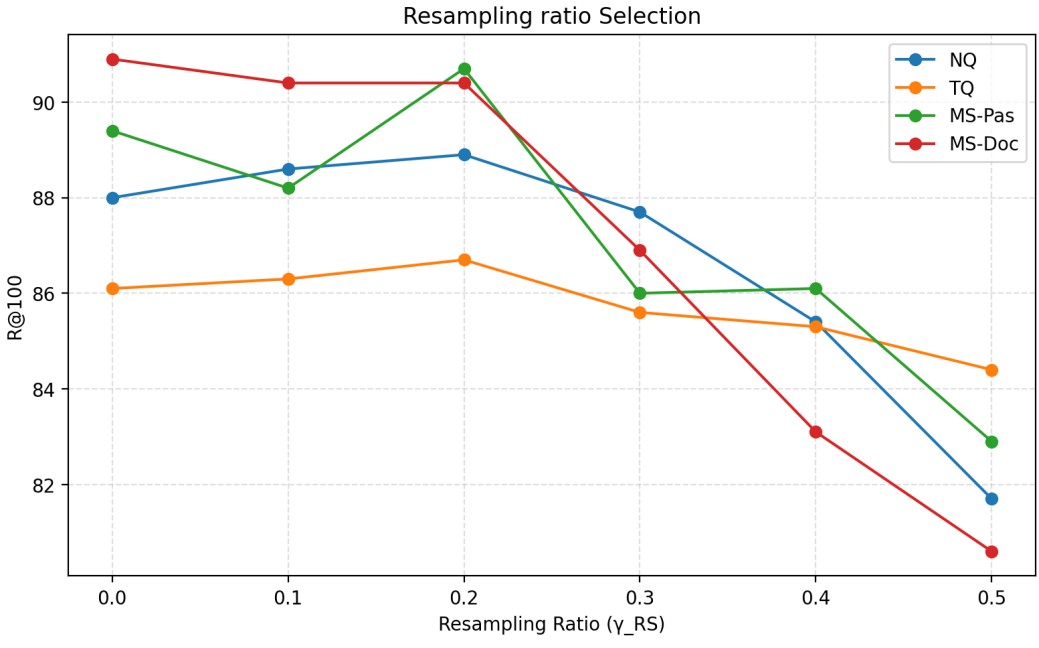

Figure 1: **R@100 vs. Resampling Ratio $\gamma_{RS}$.** Full data with `fp16`; other settings per Appendix A.1. We plot R@100 (avg) per dataset (NQ, TQ, MS-Pas, MS-Doc) to examine the sensitivity to the resampling ratio.

## A.6 RE-RANKING RATIO SELECTION EXPERIMENT

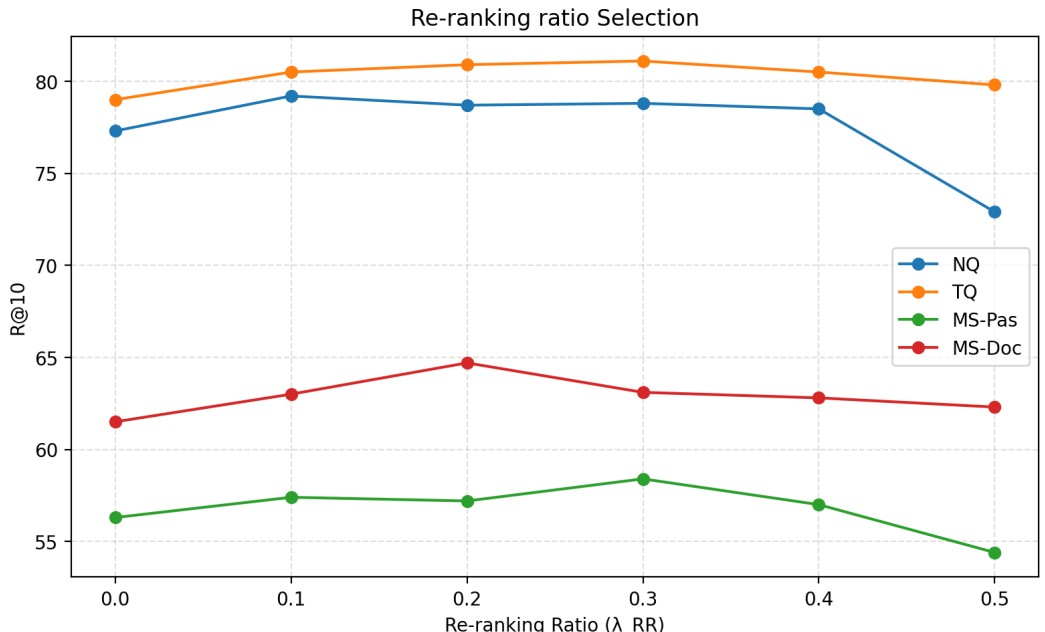

Figure 2: **R@10 vs. Re-ranking Ratio $\lambda_{RR}$.** Full data with `fp16`; other settings per Appendix A.1. To validate the re-ranking effect, we evaluate at R@10 (avg) across datasets (NQ, TQ, MS-Pas, MS-Doc).

## A.7 NEGATIVE SAMPLE NUMBER SELECTION EXPERIMENT

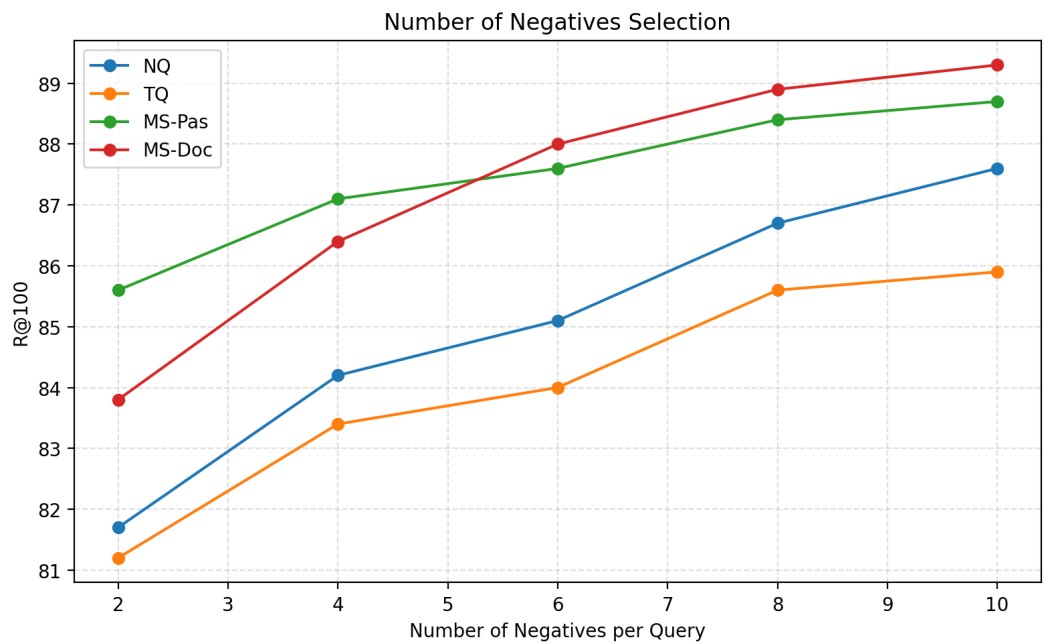

Figure 3: **R@100 vs. Number of Negatives per Query (`num_neg`).** All datasets use the full data; experiments run with `fp16` for speed; all other settings follow Appendix A.1. Curves report R@100 (avg) per dataset (NQ, TQ, MS-Pas, MS-Doc).