# OpenReview forum: "RRRA: Resampling and Reranking through a Retriever Adapter"
_ICLR.cc/2026/Conference — Submitted to ICLR 2026_

### Official Review · Reviewer_i6P1 · 2025-10-27

**Soundness:** 2
**Presentation:** 2
**Contribution:** 2
**Rating:** 4
**Confidence:** 4

**Summary:**

This paper proposes a learnable adapter module that monitors Bi-Encoder representations to estimate the likelihood that a hard negative is actually a false negative. The predicted scores are used in two downstream components: (1) resampling, where negatives are reweighted during training, and (2) reranking, where top-k retrieved documents are reordered at inference.

**Strengths:**

1. The presentation of the paper is detailed.
2. The research problem is an important problem in information retrieval.
3. The author's perspective on the problem is enlightening.

**Weaknesses:**

1. The presentation is relatively poor, the structure of this paper is somehow confusing. The introduction is too short, I cannot access the motivation and main contributions of this paper. Moreover, I don’t think Section 3: METHOD OVERVIEW deserves a whole section. Some details in Section 4.1 should be placed in the experimental details.
2. The baselines are not strong and up-to-date. No dense models in the era of LLMs are presented. The proposed method is only compared on the Bi-Encoder backbone, what about other dense backbones?
3. The motivation of this paper is also confusing, which makes this paper look like a stack of various technologies, rather than gathering together to solve a scientific problem.

**Questions:**

1. A clearer paper structure may help improve the paper.
2. More experiments on strong and new baselines as well as more backbones.
3. Reorganizing methods and motivations can help readers understand the contribution of this paper.

---

### Official Review · Reviewer_PjJ2 · 2025-10-31

**Soundness:** 2
**Presentation:** 1
**Contribution:** 1
**Rating:** 2
**Confidence:** 4

**Summary:**

The paper introduces RRRA, a bi-encoder retrieval framework augmented with a small “adapter” network that learns two auxiliary similarity measures: i) QA-score (query-adapter) and ii) CA-score (adapter-context) to handle **false negatives** in dense retrieval.

* The QA-score (==sim(q,a)) is treated as an “informativeness” signal used to boost promising candidates during inference (re-ranking, Section 5.2.3).
* The CA-score (==sim(a,c)) is treated as a “false-negative likelihood” used to down-weight uninformative or misleading negatives during training (re-sampling, Section 5.2.2).

The adapter (actually, only a MLP combined with a residual) is trained via pointwise BCE loss on pseudo-labels derived from the bi-encoder’s own predictions (Stage 2), and integrated through a residual correction and a normalization constraint. Experiments on NQ, TQA, and MS MARCO show **small** improvements in recall and early-rank precision compared with heuristic samplers (ANCE, SimANS, TriSampler).

**Strengths:**

* The idea of using one learned signal both to filter negatives during training and re-rank results at test time is **conceptually** interesting. The method avoids cross-encoders yet recovers part of their query-aware behavior via a residual correction and a simple re-ranking rule.
* In principle, the adapter sits on top of existing embeddings without adding cross-encoder/LLM reranker cost; it is known that distillation from these models is usually effective but incurs a costly (but one-time, offline) processing step.
* Some (small) improvements compared to baselines on some datasets.
* Decent ablations on the adapter internals. Removing residuals/linear normalization/ class-imbalance handling all degrades its quality which seems to support the design choices (although, they lack justification/intuition imo, see later).

**Weaknesses:**

My main overall concern is the clarity and readability of the paper. Although I am quite familiar with the IR field, I found the paper difficult to follow—from the motivation to the proposed approach. Many of the presented solutions seemed to lack clear motivation or intuitive explanation.
Below, I provide a more detailed discussion of specific concerns.

* Clarity: I read carefully the paper twice, and still struggled to understand the proposed solution (and especially, how it answers the problem, i.e., dealing with the issue of false negatives in dense retrieval training). Personally, I think the description of the method would require a major update to be easily understood. A lot of the claims through the paper lack justification; for instance, “we adopt pointwise BCE loss for stable supervision instead of InfoNCE” => Why? What happens if you use InfoNCE? Also, what’s the justification for the “linear normalization constraint”? Most of the design choices (“adapter”, QA-score/CA-score, etc.) lack a clear intuition.
* Stage-2 labels are derived from the bi-encoder’s own outcomes rather than from external relevance judgments (“using supervision derived from the Bi-Encoder’s prediction outcomes.”, Section 4.2); it is not super clear what it means exactly/how it should work. An explicit evaluation of FN detection quality vs ground truth (or cross-encoder surrogates) might be missing.
* Heuristic/ungrounded scoring rules: both re-sampling and re-ranking use ad-hoc formulas based on absolute differences and tunable exponents, which lack justification/intuition.
* The experimental setup feels a bit outdated; more recent works have discussed how to filter hard negatives effectively, e.g., [1,2,3,4]. In particular, [2] studies simple approaches with positive-aware hard-negative mining that are very relevant. Evaluations could also be improved: it is now standard practice to evaluate retrieval models on BEIR (for out-of-domain evaluation) or even MTEB. Improvements over existing samplers are small (1-2 pts) and often inconsistent across metrics (despite the strong claims like “consistently outperforms strong Bi-Encoder baselines”). On the broader comparison, SimANS/TriSampler still top RRRA on several metrics (Table 4, p.8).
* The authors claim that “c and a can be precomputed and indexed”: but from my understanding, the adapter residual is query-dependent (Section 4.3.1). Additionally. authors claim “minimal overhead” and “consistent robustness” without any timing or resource measurements.
* Some flaws in the writing: “on benchmarks such as DPR” (DPR is not a benchmark?); “Models such as ColBERT reduce inference cost via late interaction” (this is not true, or very weirdly formulated); it also seems the TP definition is flawed? Hyperparameters are also confusing (e.g., what’s ρ?).
* Gradient analysis: I don’t find the results very clear/convincing to be honest.

[1] Hard Negatives, Hard Lessons: Revisiting Training Data Quality for Robust Information Retrieval with LLMs
[2] NV-Retriever: Improving text embedding models with effective hard-negative mining
[3] Gecko: Versatile Text Embeddings Distilled from Large Language Models
[4] Gemini Embedding: Generalizable Embeddings from Gemini

**Questions:**

See questions in ’Weaknesses’ section; also:

* How are the QA-score and CA-score semantically different? If both are cosine similarities involving the same adapter vector, what makes sim(q,a) measure informativeness and sim(a,c) measure false-negative likelihood? Please provide intuition or empirical evidence supporting these interpretations.
* Can you provide qualitative examples where RRRA correctly down-weights false negatives or boosts true positives? This would substantiate the FN-modeling claim.
* How reproducible is the approach? Given the ambiguity in the paper, this also seems critical.

---

### Official Review · Reviewer_Vw6r · 2025-11-01

**Soundness:** 2
**Presentation:** 1
**Contribution:** 2
**Rating:** 4
**Confidence:** 3

**Summary:**

The paper proposes to address the important problem in retrieval training that mined hard negatives can sometimes be false negatives, which may harm the retriever’s learning process. To mitigate this, the authors introduce an adapter module that estimates the likelihood of a negative being false. This likelihood is then used both to reweight negatives during training and to rerank documents during inference. Experiments on NQ, TQ, and MS MARCO demonstrate the effectiveness of the proposed approach.

**Strengths:**

(1) This paper addresses an important problem in retrieval training: mining-based hard negatives can sometimes be false negatives, which may harm the retriever’s performance.

(2) The effectiveness of RRRA is demonstrated across multiple datasets, including NQ, TQ, and MS MARCO.

**Weaknesses:**

(1) Inconsistent comparison results. According to Tables 1 and 2, RRRA outperforms all baselines, including SimANS and TriSampler. However, in Table 4, RRRA performs worse than both baselines across all reported metrics. I assume that the results in Table 4 are taken directly from the original baseline papers, but it remains unclear what configuration differences lead to this discrepancy.

(2) Several parts of the paper lack sufficient detail, which makes it difficult to fully understand the proposed pipeline. For example, TN/TP/FP/FN are central to the model’s learning process, yet there is no explanation of how these labels are obtained.

(3) Missing ablation study. The paper lacks an ablation study analyzing how performance evolves across different training stages.

**Questions:**

(1) L019 "rewei-ghted" should be "reweighted".

(2) The citation format is incorrect. For example, "using vector similarity Karpukhin et al. (2020b)" of L026 should be "using vector similarity (Karpukhin et al. (2020b))".

(3) There should be no contributions or acknowledgements sections in the submission (although the content contained are the ones from the template).

(4) L78 "strong reranking" should be "strong reranking ablility".

(5) L53 The statement “ColBERT reduce inference cost” seems incorrect as ColBERT generally increases inference cost relative to standard bi-encoders.

---

### Official Review · Reviewer_XQCQ · 2025-11-02

**Soundness:** 2
**Presentation:** 3
**Contribution:** 2
**Rating:** 4
**Confidence:** 5

**Summary:**

This paper aims to select informative hard negatives in dense retrieval while minimize the inclusion of false negatives during training. The authors propose a learnable adapter module that estimates the probability that a hard negative is actually a false negative based on Bi-Encoder representations. This probability is incorporated into both negative sampling during resampling and reranking.

**Strengths:**

1. The motivation of this paper is clear.

2. The use of the learnable adapter in both training and inference-stage increases its practical value.

 3. The reported improvements across standard dense retrieval benchmarks demonstrate the effectiveness of the proposed method.

**Weaknesses:**

1. The adapter is trained in Stage 2 on a frozen Bi-Encoder, meaning both embeddings and hard negatives remain static. This may limit its ability to capture evolving training dynamics. When transitioning to Stage 3, the encoder is updated, which could misalign the adapter's decision boundary with the new representation space. The authors should examine whether this decoupled training causes suboptimal performance.

2. While the method is claimed to be lightweight, the paper does not provide a detailed analysis of the computational overhead introduced by the adapter, especially during reranking in inference. It remains unclear how much additional latency or memory this module adds compared to a standard Bi-Encoder, and whether the method still scales efficiently to large retrieval corpora.

**Questions:**

3. Open Question: Recent advances in large language models (LLMs), especially those with tool-use and web-enabled retrieval capabilities, raise an important question: to what extent is traditional dense retrieval still necessary, or could it eventually be replaced by LLM-based retrieval systems? While LLMs can now perform multi-hop reasoning, query reformulation, and even interact with search engines in real time, it remains unclear whether they can fully substitute dense bi-encoder retrieval? It would be valuable if the authors could briefly position their work in this evolving landscape and discuss whether methods like the proposed adapter are still critical in a future where LLMs increasingly handle retrieval themselves.

---

### Meta-Review · Area_Chair_5gkW · 2025-12-13

**Summary:**

This paper introduces a training strategy aimed at helping on the selection of hard negatives while training of bi-encoder retrievers, while also enabling re-ranking at test time. The proposal is conceptually relevant and well motivated: a three-step approach is proposed where bi-encoder retrievers and the adapter are each trained on their own, which is finally followed by a joint training step. The paper is well-motivated and the proposal is conceptually relevant, as noted by some reviewers, but the manuscript reads rushed and is unclear, and reviewers highlighted limitations in the empirical assessment to an extent that suggests no conclusive evidence to the authors claims.

**Reviewer Concerns:**

Reviewers raised various concerns regarding limitations and the clarity of the manuscript. The manuscript reads rushed and has several formatting issues.

**Reviewer Scores:**

The authors did not post rebuttals, and I would expect authors to mostly maintain their original scores.

---

### Decision · Program_Chairs · 2026-01-26

Reject